# Implementation of a Model-Based Programme to Promote Personal and Social Responsibility and Its Effects on Motivation, Prosocial Behaviours, Violence and Classroom Climate in Primary and Secondary Education

**DOI:** 10.3390/ijerph16214259

**Published:** 2019-11-02

**Authors:** David Manzano-Sánchez, Alfonso Valero-Valenzuela

**Affiliations:** Department of Physical Activity and Sport, CEI Campus Mare Nostrum, Universidad de Murcia, 30720 Santiago de la Ribera, Spain; avalero@um.es

**Keywords:** methodology, innovation, teaching, secondary education, primary education, curriculum

## Abstract

The present study aimed to apply a programme based on Hellison’s Teaching Personal and Social Responsibility model (TPSR), traditionally used in Physical Education, to other school subjects and analyse aspects related to motivation and satisfaction of basic psychological needs among other variables. The programme was applied for 7 months during one academic year, all students receiving at least 60% of the lessons through this teaching methodology. A mixed method research methodology and quasiexperimental design was implemented in three schools (two primary, one secondary), with a total of 29 teachers and 272 students (45 control, 227 experimental group) involved. The students completed a questionnaire before and after the study and the teachers underwent semi-structured interviews at the end of the intervention. The results indicated improvements for the experimental group in personal and social responsibility, the psychological mediator index, the self-determination index, prosocial behaviours and teacher climate, as well as a reduction in amotivation and antisocial behaviours. The results were similar for primary and secondary school. The interviews yielded positive opinions and showed suitability of the method to be applied in the rest of subjects. It is concluded that TPSR can be an appropriate methodology to be implemented in the different curriculum subjects to improve basic psychological need satisfaction, motivation, prosocial behaviours and classroom climate.

## 1. Introduction

Over the years, there has been the social demand that standard education provide students with the necessary tools to be able to adapt to society’s constant changes. A fundamental aspect in order to satisfy this demand is to promote autonomy and self-sufficiency among students [1,2] and to improve the learning atmosphere [3], reducing the dependence on the teacher and increasing cooperative work with classmates [4].

Among the most influencing variables on adherence to school and academic performance we can find motivation, which has a positive effect on school performance and grades [5], and satisfaction of basic psychological needs, which are key variables in the learning process and, therefore, in the improvement of academic grades [6]. There are two major theories regarding motivation. On one hand, the “Achievement goal theory” [7], based on the individual’s position as regards the motivations for their behaviours and, on the other hand, the “Self-determination theory” [8,9]. Together they form a macrotheory about motivation with the aim to understand human beings’ behaviour that can be applied to various contexts of different cultures. Along with this theory, Vallerand [10] developed a hierarchical model of motivation. This model contains several postulates, among which a predictive relationship stands out, which indicates that certain social factors (such as responsibility) may lead to increased satisfaction of the three basic psychological needs. This would produce an improvement in the most self-determined motivation, what could trigger new behaviours, such as an improvement in the classroom climate. 

In this regard, responsibility has shown positive relationships with motivational constructs and the satisfaction of basic psychological needs. Relationships with intrinsic motivation [11,12], causal relationships with autonomy and intrinsic motivation [13,14], self-concept and intrinsic motivation [15]. On the other hand, pro-social behaviours that consist of a set of actions that are beneficial to others in the form of sharing and helping, can be improved from early ages using an interactive learning environment [16], and decreasing the perceived and suffered violence [17].

This suggests that an education based on autonomy transfer to the student and responsibility promotion may be appropriate in order to improve motivation and other social factors, as well as academic performance [18]. It has also shown predictive capacity of satisfaction of basic psychological needs and the most self-determined motivation states [19].

Adolescence is the life period where externalising (e.g., antisocial behaviors, aggressiveness, mistreatment, violence) or internalising problems (e.g., shyness, social anxiety) begin or significantly increase, related with prosocial behaviours [20]. These prosocial behaviours can be understood as voluntary behaviours aimed to benefit others, and have a key role in the creation of positive interpersonal relationships and in the maintenance of personal and social wellness [21]. Carreres [22] found that the implementation of TPSR model on after-school sport had a positive relationship with personal and social responsibility, as well as with prosocial behaviours, producing a very positive perception by adolescents, families and trainers. TPSR is nowadays considered as an appropriate model to comply with the demands of the current education system and to achieve competences through active methodologies [23].

Violence is closely related with antisocial behaviours and has been defined as an intentioned behaviour through which damage or harm is caused [24]. Many different manifestations of violence may appear in the education context, among which the following can be highlighted: violence from teacher to student, direct physical violence between students, indirect physical violence from students, verbal violence between students, verbal violence from student to teacher, violence through the new information technology, class disruption and social exclusion [25]. Considering the above, classroom climate, understood as the establishment of satisfactory relationships that contribute to an adequate atmosphere, resulting from the teacher and students’ attitudes and their relationships [26], appears as a fundamental variable to achieve appropriate academic and social performance [27].

Previous studies in the education field that have applied the Teaching Personal and Social Responsibility model (TPSR), like the one by Caballero [28], have reported improvements in the classroom climate and prosocial behaviours, proving as one of the most powerful models for value development in the adolescence [29]. One of the major limitations detected was the short application time, limited to a few hours per week related with physical activity [30]. For the future, the creation of a universal community is suggested, where its use is extended to all contexts and teachers from all academic phases get jointly involved in order to achieve an improvement in the students’ quality of life [31,32].

The main aim of the present study was to gain knowledge on the effects of TPSR application in a general primary and secondary education context on responsibility, satisfaction of basic psychological needs, motivation, prosocial behaviours, violence and classroom climate. To do so, a mixed method was used, consisting of questionnaires to analyse the variables under study, video recordings to conduct a protocol for the “continuous professional development” (CPD) in order to ensure adequate fidelity of implementation, and interviews to assess the perceptions by teachers regarding the use of TPSR.

It was hypothesised that TPSR implementation on the different school subjects would lead to improved responsibility, psychological mediators, prosocial behaviours and classroom climate, as well as reduced amotivation, antisocial behaviours and violence. It was also hypothesised that the simultaneous implementation of this model would be positively perceived by the teachers of one same group of students.

## 2. Materials and Methods 

### 2.1. Design

Following Anguera, Camerino and Castañer [33], a mixed method approach was applied (mixed method of data triangulation), based on qualitative (observational analysis during the model implementation and semi-structured interviews) and quantitative instruments (pre- and post-test using a non-equivalent control group: teacher’s self-evaluation, students’ self-evaluation and questionnaires), providing continuous training and advice to teachers.

Informed consents (confidential data treatment, participation in the study and session recording) were requested from the students and their parents. An introduction letter was addressed to the two primary and one secondary schools and the approval by the ethics committee of the University of Murcia was obtained (1685/2017). The intervention programme lasted for 7 months, from the end of the first academic quarter along the second and third quarters. The contents were selected according to the current education laws. Before and after the intervention, the students answered a questionnaire in two sessions (two lessons on two different days to prevent bias due to the time limitation for questionnaire completion) in a quiet environment during 30 min per session. First, students watched a power point presentation about how to complete the questionnaires, after that the teacher read the questions in order to ensure of their understanding. The teacher and one of the researchers stood all time with them solving possible doubt. The participants were requested to provide true answers. Furthermore, a semi-structured interview was conducted at the end of the intervention with some of the teachers involved in the programme.

### 2.2. Participants

Participants were selected from three different public schools from Murcia centre and another town near to Murcia with a similar middle socio-economic level (two primary and one secondary) based on accessibility and convenience. Twenty nine teachers (20 from a primary, nine from the secondary school) were selected, 13 women (11 from a primary school and two from the secondary school) and 16 men (nine from a primary school and seven from the secondary school), with ages ranging between 28 and 49 years old.

The sample of students originally consisted of 395 participants. The following exclusion criteria were established: (a) to complete all test scales, (b) to complete the pre- and post-tests on both occasions, and (c) to complete at least 90% of the test items (excluding double answers). After applying the exclusion criteria and calculating Mahalanobis distance to remove outliers, the final sample consisted of 272 participants (207 from a primary and 65 from the secondary school, 45 of the latter constituting the control group). The sample was composed of 133 boys and 139 girls (92 boys and 114 girls in a primary school and 40 boys and 25 girls in the secondary school), with ages ranging between 9 and 14 years old (from 4th year of primary school to 3rd year of secondary school), the mean age being 11.13 (SD = 1.78).

The distribution of age and gender was similar in the control and experimental groups. None of the participants (teachers or students) had had previous experience with TPSR. Teachers’ behaviours were monitored through video recording and analysis by analysts qualified in observational methodology and TPSR.

### 2.3. Instruments

During data collection, different instruments were used for students and teachers.

#### 2.3.1. Questionnaires for Students

A closed-question questionnaire was used in the present study (attached as a Appendix A) It was divided into two parts: the first one, for sociodemographic variables, and the second one, which included the different scales used in the study. All scales had a reliability process in order to ensure their adequacy for primary, secondary and both students together. Originally, all scales (except the EME one) were validated in a student sample. Students completed the first part of the questionnaire with 64 items in one moment, and the second part with 84 or 74 items (secondary or primary) in a second time. The questionnaire as a whole had a reliability of 0.898 (pre-test) and 0.939 (pos-test).

(1) Personal and Social Responsibility Questionnaire (PSRQ): to measure personal and social responsibility levels. It was adapted to the school context by Li et al. [34] and for Spanish by Escartí et al. [12] and validated in a 9 to 15 years old sample. This scale consists of 14 items, seven to assess social responsibility (e.g., “I help others”) and seven for personal responsibility (e.g., “I set goals”). The answers were provided on a Likert-type scale ranging from 1 (totally disagree) to 6 (totally agree). Reliability in the pre-test was 0.79 for social responsibility and 0.75 for personal responsibility, while it was 0.86 and 0.84, respectively, in the post-test.

(2) Psychological Need Satisfaction in Exercise (PNSE): to measure the satisfaction of the need of social competence, autonomy and relationships. The scale adapted for Spanish and to the education context by Moreno et al. [35,36] and validated in a 12–16 years old sample. This scale consists of 18 items, six to evaluate each need: competence (e.g., “I am confident to perform the most challenging exercises”), autonomy (e.g., “I believe I can make decisions during the training sessions”) and relationships with others (e.g., “I feel attached to my training mates because they accept me as I am”). These were preceded by the sentence “During my training…” and the answers were provided on a Likert-type scale ranging from 1 (False) to 6 (True). Reliability in the pre-test was 0.72 for autonomy, 0.70 for competence and 0.72 for relationships, while it was 0.80, 0.72 and 0.76, respectively, in the post-test. Moreover, the psychological mediator index (PMI) was applied to evaluate the three variables jointly, yielding an internal consistency of 0.71 in the pre-test and 0.75 in the post-test.

(3) Motivation toward Education Scale (in French, EME): to measure motivation from the most self-determined types to the most external causes and amotivation. The Spanish version of the Échelle de Motivation en Éducation [37] validated by Nuñez, Martín-Albo and Navarro [38] in adult sample was used. The questionnaire passed a reliability test in order to check the understanding of the student sample in the same way as the others. The questionnaire consists of seven subscales, called intrinsic motivation to know (e.g., “because I feel pleasure and satisfaction when I learn new things”), to accomplish (e.g., “for the pleasure I feel when I improve my academic performance”) and to experience sensations (e.g., “because reading about topics I find interesting stimulates me”), identified regulation (e.g., “because it will allow me to access to the job market in my preferred field”), introjected motivation (e.g., “to prove myself that I am an intelligent person”), external motivation (e.g., “to get a more prestigious job”) and amotivation (e.g., “I don’t know, I don’t understand what I’m doing in high school”). The instrument is composed of 28 items preceded by the sentence “I go to school/high school because…”, with a seven-point Likert-type scale, from 1 (totally disagree) to 7 (totally agree) and distributed into seven subscales, five of them containing four items and two of them containing three items. The internal consistency analysis yielded the following values in the pre-test: 0.78 for intrinsic motivation to know (0.84 post-test), 0.72 to experience (0.72 post-test), 0.78 to accomplish (0.84 post-test), 0.87 for general intrinsic motivation (0.88 post-test), 0.75 for identified regulation (0.72 post-test), 0.74 for external regulation (0.73 post-test), 0.74 for introjected regulation (0.74 post-test) and 0.72 for amotivation (0.77 post-test). Moreover, the self-determination index (SDI) was applied using the formula ((intrinsic motivation × 2 + identified regulation) − (introjected regulation + external regulation)/2 − (amotivation × 2)), yielding an internal consistency of 0.85 in the pre-test and 0.87 in the post-test.

(4) Teenage Inventory of Social Skills (TISS): to assess prosocial and antisocial behaviours. It was designed by Inderbitzen and Foster [39], translated into Spanish by Inglés et al. [40] and validated in a 12–16 years old sample. The questionnaire consists of two subscales: prosocial values including positive social behaviours such as cooperation, community participation, altruism or ability to express feelings (e.g., “I offer my classmates help to do their homework”), and antisocial values such as aggressiveness, low self-esteem, social anxiety, presumptuousness or insolence (e.g., “I forget to give back things that others have lent me”). A six-point Likert-type scale is used, from 1 (does not describe me at all) to 6 (describes me totally). Internal consistency values in the pre-test were 0.81 for social behaviour and 0.88 for antisocial behaviour, while they were 0.87 and 0.81, respectively, in the post-test.

(5) Questionnaire of School Violence (Cuestionario de Violencia Escolar, CUVE): to evaluate violence perception. It was designed by Álvarez et al. [41]. There is a version for secondary school composed of eight subscales and one for primary school consisting of seven subscales. It was adapted to Spanish and to primary and secondary education contexts by Álvarez, Pérez and González [42]. Answers are provided on a Likert-type scale ranging from 1 (totally disagree) to 5 (totally agree). Total internal consistency of the questionnaire for primary school was 0.94 in the pre-test and 0.96 in the post-test, while it was 0.94 in the pre-test and 0.98 in the post-test for secondary school. Then mean internal consistency was 0.94 in the pre-test and 0.97 in the post-test.

(6) Questionnaire to assess social school climate (CECSCE): to evaluate the climate perceived by the students in regard to their class, teacher and school. It was designed by Trianes et al. [43], and validated in a 12–14 years old sample. The questionnaire consists of two subscales called “Climate relative to the school” (e.g., “Students are really willing to learn”), made up of eight items, and “Climate relative to the teaching staff” (e.g., “Teachers of this school are friendly to students”), composed of six items. A five-point Likert-type scale was used, ranging from 1 (totally disagree) to 5 (totally agree). The internal consistency analysis yielded a value of 0.83 for school climate and 0.70 for teacher climate in the pre-test, while it was 0.84 and 0.76, respectively, in the post-test. The internal consistency for general classroom climate (average of the two types of climate) yielded values of 0.71 in the pre-test and 0.85 in the post-test.

#### 2.3.2. Instruments for Teachers

Tool for assessing responsibility-based education (TARE 2.0). The Spanish version, validated by Escartí et al. [44], was used to evaluate responsibility-based teaching strategies. It consists of four sections: strategies applied by teachers to promote responsibility, strategies to develop responsibility, student responsibility during the session and general comments during the session. Only the first part was included in the present study. It consists in an observational method that uses interval data collection with the aim to record the strategies applied by teachers who are implementing the programme to promote responsibility [44]. Two observers analysed the presence or absence of the categories described during 3-min periods. The 29 teachers were filmed during at least four sessions (one per responsibility level) and they received a behaviour report and suggestions for improvement. Four of them were randomly selected to undergo a weekly analysis in order to assess the general class evolution.

Interview. 20 teachers from the three schools volunteered to undergo individual structured interviews at the end of the programme implementation in order to assess the model effectiveness. The questions were: (1) What is your opinion on the Teaching Personal and Social Responsibility model compared to your previous lessons?; (2) Would you like to continue applying TPSR to your lessons in the future?; (3) Do you think TPSR has helped you teach your students values such as effort, respect or leadership, besides the standard academic contents?

### 2.4. Procedure

#### TPSR Intervention Programme

The sessions followed the format proposed by Hellison [45], but was modified to keep four of its five parts: (1) Initial greeting: the teacher interacted with the students to create bonds with them, (2) Sensitivity talk: the teacher presented the academic and value goals of the session, depending on the responsibility model level, (3) Activity plan: this was the greatest part of the practical lesson, where responsibility strategies were included in the different tasks, (4) Group meeting and self-assessment: at the end of every session, teacher and students shared their perceptions with regard to individual and collective responsibility and behaviours, as well as the teacher’s behaviour, pointing their thumbs up (positive evaluation), to one side (medium) or down (negative evaluation). Teachers used general (e.g., to assign tasks, to give opportunities for success, to define roles) and specific (e.g., reciprocal teaching, cooperative groups, personal working plan) strategies when implementing TPSR that were adapted to the contents taught, as shown in Table 1. Furthermore, these strategies were also applied in order to solve individual (e.g., five days clean) and collective conflicts (e.g., grandma’s law), fully integrating TPSR in all lessons, not only Physical Education [44].

### 2.5. Intervention Programme prior to TPSR

The sessions implemented in the different school subjects prior to TPSR application focused on concept development, leaving critical thinking, reflection and idea confrontation about social aspects aside [46]. Teachers focused on developing the scientific aspect of every subject [47], becoming the centre of the learning process and the ones to make decisions [48]. In the case of languages, they restricted themselves to explaining the grammar rules trying to make students learn them in a very theoretical manner and through analytic exercises, leaving aside aspects like extralinguistic factors, which foster learning [49]. In summary, the teaching strategies treated the student as a mere recipient and did not address any aspect of the academic contents apart from the purely cognitive ones.

### 2.6. Teacher Specific Professional Development

Specific professional development of teachers is needed in order to implement any education programme [50]. Teachers were trained on TPSR using a two-phase approach: (1) 5-hour course on the model theory and practice, where they were explained how to design climates in the classroom based on the model and they were provided with global and specific strategies for responsibility development. Firstly, they received the theoretical foundations of TPSR Model, the lesson structure, the five different levels of responsibility, the general and specific strategies for teaching responsibility, and the strategies for solving problems. Secondly, teachers received, acting like students, a practical lesson based on the TPSR Model. Teachers were split up into two groups, one of them was made up of physical education teachers (12 teachers) and implemented a practical lesson in a sports court. The other group (17 teachers) was made up of those teachers who taught other subjects such as Mathematics, Literature, Spanish Language, Historic, etc. and implemented a lesson in a classroom. The main changes done in the group of teachers in the classroom were: (i) part 3 and 4 of the lesson structure were joined, and (ii) some new strategies were incorporated to improve teaching responsibility levels in the classroom. For example, to promote level 2, the “petals blackboard” strategy was created, where a flower without petals is drawn on the blackboard and students must complete the class activities to achieve the petals, making a count at the end of the lesson to show the values of participation and effort reached. They were also provided a “model guide” to review the different strategies discussed, as well as other strategies [51]. (2) Continuous training: 29 teachers who were interested in following the TPSR study, had signed a consent form and their respective schools had a consent form signed about image privacy of students and were enrolled throughout the during the programme implementation (7 months). The main researcher met the teachers on several occasions and through different means. In the first week, teachers had to deliver a document where they described the structure of one of their own sessions applying the first-level model, with the activities and strategies used. The main researcher provided comments. In the second week, sessions were implemented in the different subjects and one session was filmed and evaluated by the research team, who in week 3 provided a report of the session with suggestions for improvement. This sequence was repeated along the whole intervention, as well as quarterly meetings for teachers to discuss the intervention [51]. The aim was to create a classroom climate that promoted responsibility through TPSR implementation and to develop an appropriate intervention. Students learnt responsibility in a progressive manner, moving along the different levels [44]. Nevertheless, level 5 was involved from the beginning in an attempt to transfer it to the students’ life, although specific strategies were included to address it specifically at the end of the intervention programme. Moreover, at the end of every session, teachers performed a self-assessment using TARE instrument [52], answering on a dichotomic yes/no scale, in order to promote reflection and critical analysis on the programme implementation.

The observer indicated whether the TARE categories were applied during the lessons with satisfactory results or not [44], i.e., more than 80% of all elements listed for every session. After the session analysis process, teachers were provided with comments to improve different aspects when needed. All teachers received feedback at least once per responsibility level, and four of them were randomly selected to receive this type of feedback once a week. Thus, a combination of various strategies (training seminars, video analysis, feedback and query solving cycles) was applied in order to provide adequate orientation and support to teachers before and during the whole research project [53]. At the end of every session, teachers were encouraged to evaluate their own performance during the session using TARE in order to foster reflection on the model implementation.

### 2.7. Fidelity of Implementation

Hastie and Casey [54] suggested that researchers should provide: (a) a rich description of the curricular elements of the unit, (b) a detailed validation of the model implementation based on the pedagogical model, and (c) a detailed description of the programme context to help the reader understand the research design and the results obtained. Elements (a) and (c) have already been described in the teacher specific professional development section. For the detailed validation of the model implementation, the research group filmed one in every six sessions (5 sessions, 275 min), which was analysed by external observers (5 sessions per teacher divided into 11 observation periods of 5 min). The camera was installed in the classroom six sessions prior to the beginning of the study to allow students to familiarise with it and to prevent non-spontaneous behaviours. Every teacher’s behaviours were evaluated though TARE instrument (described in the instrument section). At the end of every session, the teacher also assessed their own behaviour using TARE [52], in order to encourage reflection on the TPSR programme implementation. The observational analysis was conducted by two university students (not related with the study) with knowledge in the education research field.

The observers were trained using the following sequence established by Wright and Craig [52]. First, explanation and clarification of the meaning of each of the categories of the tool (they were put in different situation examples to distinguish them clearly). Second, the observers together visualised two completed classes implementing the TPSR (corresponding to two lessons applied in different school not related to the present study) using TARE 2.0. Third, the results of the observers were shared to unify criteria. Fourth, when observers obtained an inter-reliability upper of 80%, that meant the reliability was guaranteed, observers were ready to start the analysis of the study sessions. Total agreement (TA) was calculated through the formula: number of total agreements (NTA) divided by agreements (A) plus disagreements (D) (TA = NTA/A + D) [55].

### 2.8. Data Analysis

First, the scores’ normal distribution was verified using the Shapiro Wilk test (*p* > 0.05). Second, initial homogeneity among groups was assessed using a multivariate analysis of variance (MANOVA). Third, a repeated-measures analysis of covariance (ANCOVA) was performed to examine the effectiveness of the implementation of the intervention programmes (interaction group × time). Finally, a multivariate analysis of covariance (MANCOVA) was conducted to compare the post-test scores between groups (inter-group). The degree of change in every group was obtained through the size effect using Cohen’s formula [56]. Values > 0.1 were considered as small effect, up to 0.5 as medium, up to 0.8 as large and > 0.8 as very large effect. Furthermore, frequency and percentage analyses were conducted to evaluate the teacher’s behaviour, as well as a description of the interviews through qualitative analysis with the aim to get an internal perspective of the experience at the end of the process [57]. The software IBM SPSS 22.0 (SPSS Inc. Chicago, IL, USA) and Excel 2010 (Microsoft, Redmond, DC, USA) was used to analyse the data from the questionnaires and the observation instrument, respectively.

## 3. Results

### 3.1. Strategies Used by Teachers to Promote Responsibility

To evaluate the instruction and treatment validity, the use of strategies to promote responsibility by the 29 teachers were assessed, and the Likert scale value (0–4) of the nine teacher categories measured by TARE 2.0 was assessed. The descriptive analysis reflected values above zero in all the variables studied. On the other hand, the four teachers who were randomly selected to undergo a weekly analysis in order to assess the general class evolution always had values above zero and greater than one except for the strategies ‘transfer’ and ‘leadership’ for one of the participants.

### 3.2. Normality Analysis

Initial homogeneity among groups was assessed using a MANOVA using the pre-test in control and experimental groups. Not statistically significant differences were found at the multivariate level for groups (Wilks’ = 0.917, F_(1.536)_ = 14, *p* < 0.093). The subsequent ANOVAs did not show statistically significant differences, except in violence (F_(3.432)_ = 7.979, *p* < 0.05; η^2^ = 0.029) and antisocial behaviour (F_(4.172)_ = 6.937, *p* < 0.05; η^2^ = 0.025).

### 3.3. Inferential Analysis

To assess the effects of the intervention programme on each group, a repeated-measures ANCOVA was conducted. The interaction effect (group × time) showed that the intervention had a significant effect, since significant changes were observed in favour of the experimental group (Table 2). In particular, in intrinsic motivation to know (*p* < 0.05), intrinsic motivation to experience (*p* < 0.01), intrinsic motivation to accomplish (*p* < 0.01), intrinsic motivation (*p* < 0.01), identified regulation (*p* < 0.01), amotivation (*p* < 0.01), autonomy (*p* < 0.01), competence ( *p* < 0.01), teacher climate (*p* < 0.01), school climate (*p* < 0.01), social responsibility (*p* < 0.01), personal responsibility (*p* < 0.01), antisocial behaviours (*p* < 0.01), social behaviours (*p* < 0.01), SDI (*p* < 0.01), PMI (*p* < 0.01) and classroom climate (*p* < 0.01). All variables increased except amotivation, antisocial behaviours (significantly) and violence. The control group only increased significantly in amotivation (*p* < 0.05) and reduced the intrinsic motivation to know (*p* < 0.05).

Finally, a multivariate analysis of covariance (MANCOVA) was conducted to compare the post-test scores between groups that allows for verification of the “impact of the programme” [58]. Results revealed statistically significant differences at the multivariate level (Wilks’Λ = 0.815, F_(19.00)_ = 3.007, *p* < 0.01), showing higher values in favour of the experimental group. The subsequent ANOVAs showed statistically significant differences with the experimental group in intrinsic motivation to know (F_(7.269)_ = 8.417, *p* < 0.01), intrinsic motivation to accomplish (F_(9.635)_ = 11.557, *p* < 0.01), intrinsic motivation (F_(5.538)_ = 5.569, *p* < 0.05), school climate (F_(2.272)_ = 6.363, *p* < 0.05), violence (F_(12.795)_ = 6.693, *p* < 0.01), antisocial behaviours (F_(12.279)_ = 6.996, *p* < 0.01), and classroom climate (F_(4.506)_ = 1.339, *p* < 0.05).

### 3.4. Interview Results

Some opinions from the teachers involved (one teacher per school) have been transcribed below from the interviews conducted.

#### 3.4.1. Opinion regarding TPSR (What is your opinion on the Teaching Personal and Social Responsibility model compared to your previous lessons?)

Teacher A (Secondary Physical Education Teacher): “It was somewhat complicated at the beginning, but 2–3 weeks later lessons became easier to give and to control, and students’ predisposition increased”.

Teacher B (Primary Tutor Teacher): “Yes, I believe it has been very positive, very useful for groups with behaviour problems. It would be advisable to keep working with this group of students, as positive results start to be perceivable. This working model brings to light the need to educate students to be socially responsible, beyond the mere knowledge acquisition, which is actually included in most of the goals for this period”.

Teacher C (Primary English Teacher): “I liked it. Up to now, the traditional way of teaching has had results on the personality development that we cannot measure empirically. It is convenient to reconsider the current methodology in order to improve results and contribute to more responsible and humanist citizens”.

#### 3.4.2. Possibility of keeping TPSR (Would you like to continue applying TPSR to your lessons in the future?)

Teacher A (Secondary Physical Education Teacher): “I really believe that this methodology could be used in the future, regardless of the results being used for research or not; it is very easy and especially when you get used to it”.

Teacher B (Primary Tutor Teacher): “Of course, I would like to continue using the model methodology in the future. I consider it a long-term approach, in which I still need to learn a lot. I have achieved some of the goals established and I feel very satisfied, but I am convinced that, with some training on the model, I will obtain many benefits for my students and myself”.

Teacher C (Primary English Teacher): “I would like it and it is necessary, as well as its promotion among our peers”.

#### 3.4.3. Value acquisition through TPSR (Do you think TPSR has helped you teach your students values such as effort, respect or leadership, besides the standard academic contents?)

Teacher A (Secondary Physical Education Teacher): “I think it has helped in this regard but, since we only reached level 3 due to the time limitations, leadership could not be addressed and insufficient time was dedicated to autonomy”.

Teacher B (Primary Tutor Teacher): “Yes, all those who attend regularly have improved. Depending on the case, some have improved in all aspects and others, despite not having improved in effort or autonomy, have improved in respect”.

Teacher C (Primary English Teacher): “Yes, those are actually the values that get most attention through this model and improvements can be perceived in the short term”.

## 4. Discussion

The main aim of the present study was to gain knowledge on the effects of TPSR application in a general primary and secondary education context following a continuous professional development (CPD) strategy on responsibility, basic psychological need satisfaction, motivation, prosocial and antisocial behaviours, violence and classroom climate. A mixed research method was used, based on questionnaires to measure the variables under study, interviews to assess teachers’ perception and video recording to conduct the continuous training and to ensure fidelity of implementation.

The importance of a sustained TPSR implementation, as well as the establishment of clear guidelines for its correct use, make the concept of “continuous professional development” (CPD) [59] an essential aspect to guarantee success in teachers’ training and in the achievement of the expected results by students. In the present study, this has been kept in mind by applying systematic observation, in particular, the “Tool for assessing responsibility-based education” [52], very positive for both CPD [60] and research [61], together with other strategies such as daily teacher evaluations, interviews and daily query solving with the main researchers.

Regarding questionnaires, responsibility improved significantly at personal and social levels, what has been proved by other authors who applied TPSR, like Sánchez-Alcaraz et al. [62]. They found improvements in both variables, what may also reveal that the model duration was appropriate, since authors like Llopiz-Goig et al. [30] indicated that improvements in personal responsibility appeared when the model was applied for a long time.

As regards the satisfaction of basic psychological needs, it is noteworthy that the psychological mediator index (combination of the satisfaction of the three needs: autonomy, competence and social relationships) improved significantly after the model application. Not many studies have addressed the relationship between TPSR and the three basic psychological needs, but it must be highlighted that in the study by Merino-Barrero et al. [18] the responsibility perceived by students predicted the satisfaction of the three basic psychological needs and the most self-determined motivation. Furthermore, Menéndez and Fernández [63], who applied a hybrid model of TPSR and Sport Education, achieved very positive results, especially in competence and social relationships. Positive results were also obtained through the combination of the method with Cooperative Learning [64]. Moreover, Manzano-Sánchez and Valero-Valenzuela [65] found a positive relationship with the third basic psychological need, autonomy, as reported in the present study.

The experimental group improved significantly in general intrinsic motivation and identified regulation, i.e., the constructs that are most closely related with self-determined motivation. This was also shown by the pre- to post-test differences in the Self-Determination Index. Again, not many studies have analysed the application of TPSR related to motivation in an education context. However, relationships have been reported between the increase in responsibility and the most self-determined motivation [11,12,15,17], as well as between the satisfaction of the three basic psychological needs and the most self-determined motivation [18], confirming the results of the present study.

The study by Caballero [28] is in agreement with the present study as regards prosocial behaviours, which improved significantly after applying the model. Moreover, Carreres [21] discovered that TPSR produced an improvement not only in personal and social responsibility, but also in prosocial behaviours, these results being in line with the present study as well.

Antisocial behaviours decreased significantly in the experimental group compared with the pre-test. Despite the fact that violence perception did not show any significant improvement, it did improve slightly compared with the pre-test. The opposite occurred with the control group, suggesting that the model could also help reduce this perception by children and adolescents, although further research is needed in this regard. Sánchez-Alcaraz et al. [66] obtained similar results in a study where they analysed the evolution of these variables after TPSR implementation.

Very positive results have also been obtained with regard to classroom climate. Improvements between the pre- and post-test have been found in school climate and teacher climate, confirming the study by Caballero [28], who reported improvements in the classroom climate through TPSR application in outdoor activities.

The second method applied in the present study, the interviews to the teachers involved, yielded very positive opinions, highlighting its simplicity after getting used to it, its applicability to the different school subjects and its usefulness to improve the students’ behaviour. All teachers agreed that they would like to continue applying the model in the future. This is in keeping with the perception by primary school teachers in the study by Manzano-Sánchez and Valero-Valenzuela [65], where it was concluded that TPSR could be extended to other subjects or education areas. In this regard, and in accordance with the present study, Hortigüela et al. [67] applied TPSR with university students of Early Childhood, Primary or Physical Education in Spain, Costa Rica and Chile and observed a very positive perception by the students of the pedagogical possibilities of TPSR in their future careers as teachers. Consequently, our study disagrees with the one by Martos et al. [68], where teachers perceived the innovative methodologies to promote autonomy and responsibility as complex to be applied due to limited time available.

The present study, given the scope of TPSR application, could serve as reference for future studies in the education field. In our case, although very positive results have been obtained in the analysed variables, it has only been possible to compare them with studies in Physical Education, after-school sport activities or other sport activities. Thus, the idea of Martinek and Hellison [31] of creating a community for the interdisciplinary programme development with teachers of the same school year, social bodies and especially in new contexts, like it was done in the present study, is very interesting. Furthermore, the use of the mixed methodology helps to extract conclusions taking into account various points of view and perspectives.

One of the major limitations that it must be mention is the sample selection was based on accessibility and convenience. The loss of sample was noteworthy, even after conducting the questionnaires twice to prevent random answers. Lastly, the lack of control in the sessions conducted with the control group may have partly led to potential bias.

## 5. Conclusions

The application of TPSR with a sample of primary and secondary school students produced general improvements in responsibility, basic psychological need satisfaction, the most self-determined motivation, prosocial behaviours and classroom climate. It also led to a positive trend in antisocial behaviours and perceived violence. TPSR implementation in all academic subjects covering at least 60% of the teaching time can be applied to both primary and secondary education with very positive results. Teachers who applied TPSR in their subjects perceived the model as an appropriate methodology to improve values and behaviours, easy to apply and feasible to be maintained in the future.

The experience based on continuous professional development (CPD) allows teachers to apply existing research, getting continuous monitorisation and gaining security, what leads to more adequate training and better intervention management. Future research lines could replicate this study in other school contexts using a similar methodology, combining qualitative and quantitative instruments, as well as including other variables of interest following Vallerand’s sequence. Furthermore, it could be interesting to examine whether causal relationships exist among basic psychological need satisfaction, motivation and variables analysed in the present study such as social factors, where responsibility could be considered as a main cause.

## Figures and Tables

**Table 1 ijerph-16-04259-t001:** Lessons, responsibility levels, strategies, contents, and task examples among the implementation.

Level of Task Example	Secondary Teacher (Physical Education)	Secondary Teacher (History)	Primary Teacher (Language)	Primary Teacher (Physical Education)
Subject-matter	Fitness: tests, strength, endurance, speed and mobility	Prehistory: Paleolithic, Neolithic and metal age	Vocabulary: types of dictionary.	Cooperative-Opposition games
			Spelling: accentuation rules	
			Grammar: text,	
			paragraph, sentence	
			and word	
	Latin dancing: salsa and merengue	Old Age: Egypt, Greece and Rome	Vocabulary: synonyms and antonyms	Volleyball: technique and tactic
			Spelling: accentuation	Physical condition: test and comparison of outcomes
			Grammar: syntax	
Task example Level 1	Circuit training: in groups of 4–5 people. They have to do a number of repetitions in every station, student may do them or not but at least they have to go together.	Historic timeline: in groups 4–5 people. They have to draw a timeline with the events that occurred during the studied periods, giving to the students the choice not to participate but respecting the rest of the mates.	Literature: in small groups of 5 people, read the book “The Little Prince”. Every student has to write the character with he/she feels more represented, making a story among all of them and telling the rest of the groups. Those who do not want to participate can only write their character.	Dodge ball game, with two fields and three cemeteries. Students who do not want to play, can be settled in the central cemetery to retrieve balls that go out and leave them in the centre to be taken by the fastest player.
Task example Level 2	Creating a choreography: students have to create a merengue choreography where everyone has to contribute with an individual step and participate in the group choreography.	Punic Wars: from an event list, students have to answer as many questions as they can individually.	Syntax: each student receives a list with 10 syntax problems, in progression of complexity. Each student tries to solve all that he/she can receiving a point or a reward for each sentence he/she gets to do rightly	A volleyball reduced game: they have to play a 4 vs 4 match and they have in a Borg scale (1–10) to up 8 points.
Task example Level 3	Fitness: in small groups of 4 students, have to expose to the rest of the classmates a progression routine to improve the speed, strength or endurance	The Great Battles. Students have to do an individual task where they look for information about and battle history, origin, main characters, current consequences and personal conclusions to present at the end of the learning unit to their classmates	Spelling: accentuation rules. Individually, each student has to look for on internet typical word from Murcia Region, indicating if they have the stress in the final, second-to-last or third-to-last syllable. Verbal explanation to classmates of the meaning of these words.	Individual work plan. Students after doing Alpha Fitness Children Battery and comparing their outcomes with the average values, they Will elaborate an individual work plan with 5 sessions to improve the physical ability the most like and with that with the lowest outcome
Task example Level 4	Fitness: groups of 5 students have to create their own circuit training with 4 stations to improve their strength. One student of the group Will be responsible for choosing the next station to go and finally, he/she explains to the rest of the groups what they have done in every one of the four stations	History of Rome: groups of 4 students, each group does its own work on the History of Rome for 5 lessons. Each lesson will have a leader who will be responsible for writing the report to be delivered to the teacher at the end of each class	Spelling: groups of 5 students play the contest “Up the pencil”. The teacher says a letter and a family of words (for example A and fruits). Each group collects as many words as possible and the leader of every group chooses only those ones which are right. When the teacher gives the final sign every leader will say all the words of his/her group had collected.	Cooperative-opposition games: groups of 4 students have to play an alternative games (for example “colpbol”. The most skilled players will help the rest of the team to get a goal.
Task example Level 5	Latin dancing: workshop for family students to teach a Latin dance choreography to their parents including some steps they have learnt previously during the physical education lessons.	Ancient world: An ancient theatre. Students are invited to participate in a theatre play about Punic Wars where, they can choose between being audience or actors and actresses.	Accentuation: after working accentuation and grammar rules, the game “goose of the letters” is carried out inviting the 4th grade students planning a human goose in teams with 4th level Language questions. Each 4th grade Student is accompanied by a 6th grade Student who help him but never say the answer.	Cooperative games: the 6th grade students after finishing the cooperative games unit, in the party at the end of the term, they invite the 4th students to participate in a game session lead for them.

**Table 2 ijerph-16-04259-t002:** Analysis of the intervention results.

Variables of Study	Group	Pre-Test		Post-Test		Pre-Post Difference		Inter-Group Difference of Means	
		Mean	SD	Mean	SD	*p*-Value	ŋ^2^	Dif	*p*-Value
IM to know	Control	5.82	1.05	5.54	1.23	0.028 *	0.24	−0.28	0.006 **
	Experimental	5.85	1.05	6.01	1.04	0.014 *	0.15	0.16	
	*p*-Value + ŋ^2^	0.859	0.03	0.007 *	0.44				
IM to experience	Control	5.25	1.25	5.18	1.17	0.632	0.06	−0.07	0.145
	Experimental	5.10	1.25	5.31	1.18	0.010 *	0.17	0.21	
	*p*-Value + ŋ^2^	0.452	0.12	0.509	0.11				
IM to accomplish	Control	5.62	1.27	5.49	1.40	0.480	0.10	−0.13	0.041 *
	Experimental	5.79	1.13	6.05	1.03	0.001 **	0.24	0.26	
	*p*-Value + ŋ^2^	0.346	0.15	0.002 **	0.51				
General IM	Control	5.56	1.05	5.40	1.18	0.186	0.14	−0.16	0.016 *
	Experimental	5.58	1.02	5.79	0.96	0.001 **	0.21	0.21	
	*p*-Value + ŋ^2^	0.914	0.02	0.019*	0.39				
Identified R.	Control	5.85	0.94	5.73	1.12	0.388	0.12	−0.12	0.029 *
	Experimental	5.72	1.15	6.01	1.00	0.000 **	0.27	0.29	
	*p*-Value + ŋ^2^	0.404	0.12	0.090	0.27				
Introjected R.	Control	5.39	1.25	5.52	1.44	0.407	0.10	0.13	0.839
	Experimental	5.62	1.14	5.71	1.11	0.248	0.08	0.09	
	*p*-Value + ŋ^2^	0.214	0.20	0.392	0.16				
External R.	Control	5.83	1.27	5.74	1.10	0.569	0.08	−0.09	0.557
	Experimental	5.88	1.11	5.90	1.09	0.794	0.02	0.02	
	*p*-Value + ŋ^2^	0.792	0.04	0.354	0.15				
Amotivation	Control	1.65	1.15	1.31	0.54	0.049 *	0.38	−0.34	0.385
	Experimental	1.51	0.99	1.33	0.65	0.009 **	0.21	−0.22	
	*p*-Value + ŋ^2^	0.416	0.14	0.867	0.03				
Autonomy	Control	3.73	0.68	3.82	0.91	0.513	0.11	−0.09	0.221
	Experimental	3.55	0.82	3.81	0.75	0.000 **	0.33	0.26	
	*p*-Value + ŋ^2^	0.165	0.23	0.948	0.01				
Competence	Control	4.15	0.68	4.16	0.77	0.959	0.01	0.01	0.101
	Experimental	4.07	0.63	4.24	0.58	0.000 **	0.28	0.17	
	*p*-Value + ŋ^2^	0.466	0.13	0.498	0.13				
Social relationships	Control	4.37	0.80	4.19	0.86	0.078	0.22	−0.18	0.033 *
	Experimental	4.34	0.66	4.40	0.63	0.181	0.09	0.06	
	*p*-Value + ŋ^2^	0.825	0.04	0.055	0.31				
Teacher climate	Control	4.31	0.54	4.28	0.59	0.826	0.05	−0.03	0.360
	Experimental	4.30	0.55	4.40	0.56	0.006 **	0.18	0.10	
	*p*-Value + ŋ^2^	0.585	0.02	0.159	0.21				
School climate	Control	4.01	0.71	4.01	0.65	0.482	0.00	0.00	0.079
	Experimental	4.16	0.66	4.26	0.59	0.007 **	0.16	0.10	
	*p*-Value + ŋ^2^	0.408	0.22	0.019 *	0.42				
Social responsibility	Control	5.44	0.56	5.36	0.56	0.381	0.14	−0.08	0.021 *
	Experimental	5.37	0.68	5.49	0.58	0.000 **	0.19	0.12	
	*p*-Value + ŋ^2^	0.377	0.11	0.224	0.23				
Personal responsibility	Control	5.46	0.54	5.33	0.70	0.200	0.21	−0.13	0.005 **
	Experimental	5.35	0.65	5.53	0.59	0.000 **	0.29	0.18	
	*p*-Value + ŋ^2^	0.288	0.17	0.054	0.33				
Violence	Control	2.17	0.80	2.23	0.86	0.658	0.07	0.06	0.297
	Experimental	1.86	0.62	1.81	0.69	0.187	0.08	−0.05	
	*p*-Value + ŋ^2^	0.020 *	0.47	0.000 **	0.58				
Antisocial behaviours	Control	2.32	0.93	2.31	0.84	0.954	0.01	−0.01	0.315
	Experimental	1.99	0.74	1.88	0.74	0.000 **	0.15	−0.11	
	*p*-Value + ŋ^2^	0.009 **	0.43	0.001 **	0.57				
Prosocial behaviours	Control	4.15	0.78	4.04	0.85	0.427	0.13	−0.11	0.019 *
	Experimental	4.09	0.70	4.26	0.67	0.000 **	0.25	0.17	
	*p*-Value + ŋ^2^	0.592	0.08	0.060	00.31				
SDI	Control	8.07	3.31	8.29	2.66	0.609	0.07	0.22	0.141
	Experimental	8.10	3.36	9.13	2.98	0.000 **	0.32	1.03	
	*p*-Value + ŋ^2^	0.953	0.01	0.081	0.29				
PMI	Control	4.08	0.57	4.06	0.76	0.769	0.03	−0.02	0.033 *
	Experimental	3.99	0.56	4.15	0.52	0.000 **	0.30	0.16	
	*p*-Value + ŋ^2^	0.310	0.16	0.422	0.16				
Classroom climate	Control	4.16	0.54	4.14	0.55	0.770	0.04	−0.02	0.116
	Experimental	4.23	0.54	4.33	0.54	0.001 **	0.19	0.10	
	*p*-Value + ŋ^2^	0.430	0.13	0.040 **	0.35				

Note: * *p* < 0.05; ** *p* < 0.01; IM = Intrinsic motivation; R = Regulation; ŋ^2^ = Size effect (Cohen’s D); Dif = Difference of means.

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
