# Peer review of "Implementation of a Model-Based Programme to Promote Personal and Social Responsibility and Its Effects on Motivation, Prosocial Behaviours, Violence and Classroom Climate in Primary and Secondary Education"

_ijerph, 2019, doi:10.3390/ijerph16214259_

Round 1

Reviewer 1 Report

Overall this is an interesting paper that can make a contribution to the current research on educational interventions to improve prosocial behavior and classroom climate in primary and secondary education. This article reports the findings of a mixed-methods study with primary and secondary students in three schools in Spain. This is an interesting topic and the paper as a whole would be of interest to the journal audience. However, the paper presents some major issues to be addressed it can reach the standards to be published in IJERPH.

Firstly, the abstract presents some inconsistencies in the concepts that are introduced (i.e. “improvements in … teacher climate”??). Also, the title only refers to “prosocial behaviours, violence and classroom climate” whereas later in the paper motivation appears as an important variable to be measured, as presented in the introduction and using a particular instrument to measure it. I think the focus on motivation should be clarified.

This is followed by the introduction which presents an account of the literature that presents, actually, the literature on motivation, adolescence, violence and TPSR. Overall, this section presents relevant and updated research to provide context and explain other findings from existing research in the field. Most of the literature, however, refers to adolescence, secondary or high school students. I would suggest including some recent studies conducted with elementary students (i.e. Villardon et. al 2018 on prosocial behavior).

 My main concerns refer to the Methods. The paper mentions that “the students completed a questionnaire” divided in two parts, sociodemographic data and a series of questionnaires used. This second part includes 6 different instruments, described afterwards. However, there is no clear the total of items that students completed. It sounds like a quite high number of items for young children to reply, although it was completed in two sessions. There were students ranging from 9 to 14 years. Were these scales validated and suitable for all ages? I think some more clear information about the overall questionnaire should be provided, or actually, the instrument itself as an appendix to the paper. On the other hand, although the study presents a mixed-method design, and teachers’ qualitative data is provided, this raises a limitation, taking into account that no qualitative data from most of the participants is provided. Furthermore, qualitative data from the teachers has not been analyzed, it is simply transcribed and presented descriptively. This presents several limitations for the clarity and transparency of the study. Some of the contributions from Creswell & Clark (2017), or Mertens (2017) mixed-methods design would be useful for the authors to address this weakness.

P 6, line 57 / P 7, line 3,4: “The focus of the research was…” there is no clear if this statement refers to the research presented in the paper, or to the SBM program describe

Other relevant information missed is about the program and particularly about the mentors. Come more information about the context where the program takes place, how this is organized, who are the mentors, and if they do receive any training, would provide clear information and give more credibility to this case.

Very limited characteristics of the young people are provided. A table or extended information about them would improve the section.

Discussion reflects on the literature and the claims in this section are supported by the results. Conclusions are brief and summarize the most important findings, but there is no acknowledgement for any limitations of the study.

Hope these comments may be useful for the author(s).

Author Response

We have included all the answers to the reviewer's considerations in the attached manuscript, modifying those aspects that can be improved.
The only statement that we don't understand is:"P 6, line 57 / P 7, line 3,4: “The focus of the research was…” there is no clear if this statement refers to the research presented in the paper, or to the SBM program describe".

If the reviewer considers that it is very important, we would like if it is possible to clarify what it means to be able to modify it in the manuscript

We include an example of the questionnaire,

Thanks you.

Reviewer 2 Report

Very good study, purpose, design and outcome. This study is very significant because it is the 1st time I've seen a study that originated in physical education to be implemented into other academic areas. Usually physical education is put in the position that they have to implement something from other academic areas.

a couple concerns in the United States because many children may have IEP's that indicate all instructions are to be read to them- how were the survey's given? Were the questions read to the primary grade children? if not how did you know if they understood what they were reading.

the number of primary grade children were significantly more then secondary level so how did they effect the results

the antidotal notes from the teachers provide: what grade level was it evenly distributed across the grade?

Author Response

The answers to all the questions indicated are reflected in the manuscript. Any aspect to modify do not hesitate to ask us.
We include an example of the questionnaire,
Thank you

Round 2

Reviewer 1 Report

The authors addressed all the changes suggested that are clearly highlighted in the manuscript. This revised version manuscript fulfills the standards of the journal to be published in its present form.